# An Exploration of the Effectiveness of a Peer-Led Pain Management Program (PAP) for Nursing Home Residents with Chronic Pain and an Evaluation of Their Experiences: A Pilot Randomized Controlled Trial

**DOI:** 10.3390/ijerph17114090

**Published:** 2020-06-08

**Authors:** Mimi Tse, Yajie Li, Shuk Kwan Tang, Shamay S. M. Ng, Xue Bai, Paul H. Lee, Raymond Lo, Suey Shuk Yu Yeung

**Affiliations:** 1School of Nursing, The Hong Kong Polytechnic University, Kowloon, Hong Kong SAR, China; leah.li@polyu.edu.hk (Y.L.); sk-angel.tang@polyu.edu.hk (S.K.T.); paul.h.lee@polyu.edu.hk (P.H.L.); 2Department of Rehabilitation Sciences, The Hong Kong Polytechnic University, Kowloon, Hong Kong SAR, China; shamay.ng@polyu.edu.hk; 3Department of Applied Social Sciences, The Hong Kong Polytechnic University, Kowloon, Hong Kong SAR, China; xue.bai@polyu.edu.hk; 4Department of Geriatrics and Palliative Medicine, Shatin Hospital, Hospital Authority, Kowloon, Hong Kong SAR, China; losk@ha.org.hk; 5Department of Human Movement Sciences, Vrije Universiteit Amsterdam, 1081 HV Amsterdam, The Netherlands; s.y.yeung@vu.nl

**Keywords:** peer led, pain management program, pain education, older adults

## Abstract

*Background:* 80% of nursing home residents have reported chronic pain, which is often accepted by older adults as part of aging. Peer support models are being used to help individuals manage their chronic conditions and overcome the challenges of limited healthcare resources. The aims of this study were: (i) to examine the effectiveness of a 12 week peer-led pain management program (PAP) for nursing home residents and (ii) to evaluate their experiences. *Methods:* A cluster randomized controlled trial (RCT) was used. The 12 week pain management program was provided for the experimental group. Outcomes were measured at three time points. The participants’ satisfaction and acceptance were evaluated by a semi-structured interview after the program was completed. *Results:* Pain self-efficacy, pain intensity, pain interference, pain knowledge, and depression levels improved after the completion of the 12 week peer-led PAP. The pain-intensity level reported at week 12 was significantly lower in the experimental group than in the control group. Semi-structured interviews showed that the nursing home residents were satisfied with the pain education that they received. *Conclusions:* The 12 week peer-led PAP appeared to improve the pain-related and psychological outcome measures in nursing home residents, and the feedback on the peer-led PAP from the nursing home residents was positive.

## 1. Introduction

Chronic non-cancer pain is common among older adults, with approximately 50% of community-dwelling older adults [1] and up to 80% of nursing home residents reporting pain [2]. Chronic pain is associated with various physical and psychological incapacities [1,2,3]. However, older adults often accept chronic pain as part of aging and have concerns about possible adverse reactions to analgesics; therefore, pain is often inadequately managed [4]. Due to the low compliance with pharmacological approaches to managing pain, non-drug strategies such as pain education, exercise, and visual stimulation are becoming increasingly popular [2]. Chronic pain management programs (PAPs), including providing education about pain, the use of drugs, and the practice of various non-drug techniques, have been shown to be effective in reducing pain and improving physical and psychological parameters among nursing home residents [5].

With limited healthcare resources, the implementation of PAPs by healthcare professionals may be challenging. As such, peer-support models via the provision of emotional, informational, and relationship support are being used to help individuals manage their chronic [6,7]. A 12 week PAP significantly reduced pain intensity, enhanced the activities of daily living, and decreased the levels of loneliness among nursing home residents, regardless of whether the program was led solely by healthcare professionals or with the use of peer volunteers (PVs) [8]. However, the aforementioned study was a quasi-experimental study and whether the effectiveness of the peer-led PAP could be sustained after the end of the program was not investigated. Apart from investigating the effectiveness of a peer-led PAP, it is also worth exploring the experiences encountered by the end users. This information can lead to a better understanding of their experiences and perceptions of the peer-led PAP, and is essential for planning future programs. Previous studies have investigated the experiences from the perspective of PVs and community-dwelling older adults [8,9,10,11]. However, such information is lacking with regard to nursing home residents, a sub-population who tend to have a higher prevalence of chronic pain, who may encounter more difficulties in seeking pain management in a “closed” nursing home environment, and are generally physically frailer than community-dwelling older adults. 

The aims of this study were to examine (i) the effectiveness of a 12 week peer-led PAP for nursing home residents on pain-related and psychological outcome measures directly after and 3 months after the completion of the program as compared to the control group that received the usual care; and (ii) to evaluate the experiences of the nursing home residents.  

## 2. Methodology

### 2.1. Study Design

A cluster randomized controlled trial (RCT) was used, i.e., randomization at cluster levels (nursing home level). All of the participants from the same nursing home were randomized by a third person to either the experimental group or the control group according to a computer-generated list, at a 1:1 ratio, in order to avoid the potential of communication between the groups and the possible sharing of information. 

### 2.2. Sample and Procedure

#### 2.2.1. PVs

PVs to lead the pain education in the experimental group were recruited from an institute hosted by a local university in Hong Kong. Eligible volunteers had to satisfy the following criteria:(1)Aged > 55 years;(2)Scored > 6 in the Abbreviated Mental Test, indicating that they had the mental/cognitive capacity to serve as elderly peer volunteers;(3)Be willing to attend training workshops and biweekly meetings with the research team for case reviews, discussions, and to reinforce strategies on pain management education;(4)Pass an exit test (including a knowledge test on pain management) showing their ability to demonstrate various non-pharmacological practices and use the teaching manual (the principal investigator and one of the co-investigators were the assessors, and supplementary classes were given to those PVs who did not pass the exit test);(5)Be willing to lead the PAP in a nursing home.

After the eligible PVs were recruited, four 2 h training workshops conducted over 2 weeks were provided for the PVs to learn the related knowledge. The topics of the workshops included: (1) what a peer is; (2) communication skills; (3) client safety and confidentiality; (4) managing crises and emergencies; (5) motivational strategies to enhance the compliance of the participants; (6) demonstrations on the use of the teaching manual, i.e., “I can do it” and various non-pharmacological practices. All of the materials were uploaded to Google Drive for the PVs to review at any time. The workshops were conducted in small groups. PVs were required to pass an exit exam after the workshops. 

#### 2.2.2. Older Adults

Older adults were recruited between 2018 and 2019 from different government-subsidized nursing homes run by the Social and Welfare Department of Hong Kong. Managers of nursing homes were approached and residents who met the following criteria were invited to participate. Inclusion criteria included (i) aged > 60 years; (ii) scored > 6 in the Abbreviated Mental Test, where a cut-off point of 6 is valid for differentiating between geriatric clients with normal and abnormal cognitive functions; (iii) have been experiencing non-malignant physical pain or discomfort either all of the time or on and off for >3 months, with a pain score of ≥4 (on a 0–10-point pain scale in the Brief Pain Inventory), as studies have shown that nursing home residents have reported a moderate level of pain intensity of ≥4. To reflect the variety of pain conditions among older adults, all types of chronic non-malignant pain problems were included; (iv) scored > 60 points on the Chinese version of the Modified Barthel Index, indicating the moderate dependence in performing activities in the pain management program; and (v) able to speak and understand Cantonese. Participants were excluded if they (i) scored ≥ 8 in the Geriatric Depression Scale (GDS), an indication of symptoms of depression; (ii) had a history of psychotic disorders, making them unable to understand and follow instructions; (iii) had cancer and were currently undergoing cancer treatment; and (iv) had a condition that prevented them from participating safely in exercise (such as a fracture or if they had recently undergone surgery, suffered from an acute stroke, and so on).

The study was conducted in accordance with the Declaration of Helsinki, and the protocol was approved by the Ethics Committee of the Hong Kong Polytechnic University (Project identification codeHSEARS20171218005). Information about the study and an explanation of the ethical conduct of the study were provided to all the eligible participants and their family members, and the participants were asked to sign an informed consent form. The participants were also informed that they could withdraw from the study at any time without any adverse consequences. Figure 1 shows a flow chart of the study.

## 3. Experimental Group vs. Control Group

### 3.1. Experimental Group

A pain management program (PAP) led by PVs using a teaching manual was provided for the experimental group in the nursing homes. The program started with 20 minutes of physical exercises performed under the supervision of the PVs, followed by 30 minutes of pain management education, including information on pain situations, the impacts of pain, the use of drug and non-drug strategies for pain management, and demonstrations and return demonstrations of various non-drug pain management techniques. 

At the end of the session, the PVs helped the participants make portfolio entries about the activities of the day, to help them recall the various pain relief methods learned in each class.

### 3.2. Control Group

With reference to Ersek [12], the participants in the control group received the usual care and a pain management pamphlet distributed by the nursing home staff. We believed that reading the pamphlet could help the participants to manage their pain, but that this would be less efficacious than the PV-led PAP. 

## 4. Outcome Measures

Outcome measures were administered at three time points: (1) baseline (T0): after randomization and before starting the program; (2) post-intervention (T1): immediately after the experimental group finished the education program; and (3) follow-up (T2): one month after completing the program. A series of well designed questionnaires were used to measure the outcomes. The primary and secondary outcomes were administered at T0, T1, and T2.

### 4.1. Primary Outcome

#### Pain Self-Efficacy

The pain self-efficacy questionnaire contains 10 questions on a patient’s belief in his or her ability to accomplish daily tasks in spite of pain. The answers are rated on a seven-point Likert Scale, where 0 refers to not at all confident and six refers to completely confident. Higher scores reflect greater pain-related self-efficacy. The validity and reliability of the questionnaire have been proven [13].

### 4.2. Secondary Outcomes

#### 4.2.1. Pain Intensity and Pain Interference

The Brief Pain Inventory (BPI) is a brief questionnaire that was designed to measure the intensity of pain and impairment due to pain. The BPI consists of four questions related to pain severity and seven questions related to pain interference. The items are rated on a scale ranging from 0 (no pain at all) to 10 (pain as bad as you can imagine or interferes completely). The pain interference items deal with general activities, mood, walking ability, work, relationship with others, sleep, and enjoyment of life. A previous study had proven that the BPI has a good internal consistency and an acceptable test-retest reliability [14].

#### 4.2.2. Depression

The Geriatric Depression Scale (GDS) is a 30-item self-reported assessment used to identify depression. A total score of 0–9 is regarded as “normal,” 10–19 as “mildly depressed,” and 20–30 as “severely depressed.” The validity and reliability of the scale have been tested [15]. 

#### 4.2.3. Pain Knowledge

A total of 11 questions related to the content of the pain education were designed to assess the participants’ pain knowledge. The questions included, “What is the effect of Paracetamol?”, “Can listening to music help reduce pain?”, “Is exercise effective in pain management?”, “Is it appropriate to apply a hot/cold compress when sleeping?” The questions for assessing pain knowledge were developed and validated by the research team, including a geriatric physician consultant specialized in pain, a registered physiotherapist, and an advanced practice nurse experienced in elderly care. The total score was calculated by counting the number of correctly answered questions (which was the number of correctly answered questions /11*100), with higher scores indicating a better knowledge of pain. 

#### 4.2.4. Satisfaction and Acceptance

Semi-structured interviews were conducted after the program for both the PVs and nursing home residents. Open-ended questions were asked to assess their satisfaction with this pain management program. 

For the PVs, the questions included “Please describe your experience in leading the pain management program”, “Please share your perception of the benefits of the program”, “Please share the limitations and barriers that you encountered in teaching the pain management methods”, and “Do you have any suggestions for improving the pain management program?”. 

The questions for the older adults included, “What do you think about the program?”, “Can you share your learning experiences and feelings?”, and “Do you have any suggestions to improve the program?”. The participants in the experimental group were also asked questions about the peer volunteers, such as “What do you think about our volunteers?”, “Did you enjoy the classes with the volunteers?”, and “Did you feel satisfied with the volunteers?”.

## 5. Statistical Analysis

SPSS version 23 was used for handling and analyzing the data. Outcome variables and demographic characteristics were presented using descriptive statistics. The differences in the demographic characteristics and the outcome variables of the two groups were compared using a chi-square test. A generalized estimating equation (GEE) was used to test the changes over time. The reported significance level was set at 0.05 for a two tailed test, and *p* < 0.05 was regarded as statistically significant. Qualitative data on the contents of the interview were analyzed after each interview. The interviews were tape-recorded and then transcribed and cross-checked by the research team to ensure consistency and accuracy. 

## 6. Results

### 6.1. Demographic Characteristics

A total of 68 participants who satisfied the criteria were recruited. Thirty-six were allocated to the experimental group and 32 to the control group. The demographic characteristics of the participants are shown in Table 1. More females (73.5%) joined the study than males (26.5%). All of the participants were aged between 60 and 100. More than half of the participants were widows. More than one-third of the participants were uneducated. Forty-one percent of the participants had resided in a nursing home for 1 to 3 years, and 6% for over 10 years. Hypertension was the most commonly reported chronic disease (39.7%). No statistically significant differences in the demographic characteristics were found between the experimental and control groups.

### 6.2. Pain Self-Efficacy, Pain Intensity and Pain Interference

Data on the pain self-efficacy over time of the two groups are presented in Table 2. In both groups, pain self-efficacy improved after the PAP. With regard to the experimental group, the within-group comparison revealed a statistically significant difference at the post-intervention (T1) (*p* = 0.019) and 3-month follow-up (T2) (*p* = 0.002). As in the control group, there was a clinical improvement in the pain self-efficacy with non-significant differences in the within-group comparison. However, the difference between the two groups was not statistically significant at the post-intervention assessment (T1, *p* = 0.674). In addition, a statistically significant difference was observed in the between-group comparison at the 3-month follow-up (T2, *p* = 0.040).

The pain intensity and pain interference of the two groups are also reported in Table 2. A clinical reduction in pain intensity was found in both groups at the post-intervention assessment (T1) and at the 3-month follow-up (T2). However, no significant difference was revealed in both the between-group comparison and the within-group comparison. The scores for the pain interference were reduced in both groups as well. A significant reduction in the pain interference score was seen in the within-experimental group comparison at the post-intervention assessment (T1) when compared to the baseline (T0) (*p* = 0.002). A non-statistical difference was found between the two groups at the post-treatment (*p* = 0.710) assessment as well as at the 3-month follow-up (*p* = 0.731). 

### 6.3. Depression

There was no statistical difference in the depression score between the two groups at baseline (*p* = 0.913). After the intervention (PAP), the score for depression was reduced in both groups at the post-intervention assessment (T1), and it was lower in the experimental group (3.50 ± 2.30) than in the control group (4.04 ± 3.27). Nonetheless, the difference between the two groups was not statistically significant after the treatment (*p* = 0.704) and at the 3-month follow-up (*p* = 0.848). A small effect size was observed at different time points. Details are shown in Table 3. 

### 6.4. Pain Knowledge

The scores for pain knowledge in both the experimental group and the control group improved after the members received the pain education, and the score was higher in the experimental group (53.41 ± 21.33) than in the control group (51.38 ± 16.13) (shown in Table 4). The between-group and within-group differences were not statistically significant. 

### 6.5. Satisfaction and Acceptance

#### 6.5.1. Feedback and Comments from PVs

The PVs provided the following feedback: (a) leading the pain management program (PAP) is a meaningful experience; (b) the program is beneficial in that one can help oneself as well as others; (c) participating boosted one’s sense of self-worth. Details were reported in our previous study [9]. 

#### 6.5.2. Feedback and Comments from the Older Adults

The feedback and comments from the older adults can be organized under two categories: (1) about the program; (2) about the peer volunteers. The data are presented in Table 5 below. 

## 7. Discussion

The present study showed the results of the pilot peer-led pain management program. A total of 68 participants took part in this study. After the program, the score on pain self-efficacy had improved in both groups and a statistically significant difference was revealed between the experimental group and the control group at the 3-month follow-up. Moreover, a statistically significant difference was found in the within-group comparison in the experimental group. Pain intensity, pain interference, pain knowledge and depression all improved after the treatment, with no statistically significant differences between the two groups or within each group. The participating nursing home residents indicated that they were satisfied with this pain management program and that the program was acceptable. 

There was a significant improvement in the pain self-efficacy in the experimental group. The score for pain-efficacy continued to improve during the program. Previous studies also reported significant improvements in pain self-efficacy from participation in pain management programs. Our results are consistent with those studies [16,17]. This finding is promising, as it indicates that our peer-led pain management program is effective at improving the pain self-efficacy of patients. The participants would thus be more able to carry out their daily activities in spite of their pain situations, which will enhance their quality of life. 

It has been proven that the peer support model is an effective approach to managing chronic conditions [6,10].The support and encouragement of peers would also be a reason for the significant improvement in pain self-efficacy that was observed in the experimental group. Such support can enhance the confidence of older adults in dealing with their pain. The peers involved in the program can provide social support for older adults living in nursing homes. The feedback on our peer volunteers from the participants in the experimental group was positive. As the older adults are living in a nursing home and their families are not always able to spend time with them, the interaction with peer volunteers would enhance their emotional well-being. Our participants also indicated that they would welcome more visits from peer volunteers.

Clinical improvements were found in the pain intensity and pain interference, although the improvements were not statistically significant. A previous study showed that unrelieved pain among older adults hindered their activities of daily living, impaired their mobility, and led to falls and sleep disturbances [3]. In our present study, clinical improvements in pain intensity and pain interference were observed, which can improve the activities of daily living of older adults, especially those living in nursing homes. They would be able to practice the pain relief methods taught in the PAP by the peers, in their own free time in the nursing home environment. This will help them to develop and maintain health-enhancing habits. 

Depression was clinically improved in both groups. Nursing home residents live in a “closed” environment and are usually physically frail. Studies have shown a high prevalence of depression among older adults living in nursing homes [18,19,20]. In Hong Kong, it was reported that 7.7% of nursing home residents have been diagnosed with depression [21]. Previous studies had demonstrated the important role that social support plays in bolstering psychological well-being by providing psychological resources, and had shown that social relationships can reduce psychological distress [22,23,24]. Our peer-led pain management program provided social support for the nursing home residents. Therefore, it is not surprising that a greater improvement was observed in the depression of the participants in the experimental group than in the control group. 

After the treatment, the participants’ knowledge of pain improved and the score was higher in experimental group. The reason for this difference may be due to the different formats for providing knowledge to the two groups. The face-to-face education that the experimental group received and the opportunity that they were given to practice the knowledge in a timely manner would have left a deeper impression on them than the pamphlet given to the control group to read would have left on the latter group. Thus, our peer-led program showed the potential for effectiveness over the long term. 

Feedback from the participants was collected during the semi-structured interview to explore their satisfaction with and their acceptance of this program. Details about the comments were reported in our previous study [9]. The older adults who participated in this study showed their interest in and acceptance of both the program and the peer volunteers. The promising results provide a foundation for another study that we will conduct in the future. 

There were several limitations to this study. The outcome measures relied on self-reported data, and the participants might have provided socially desirable responses to please the researchers. However, to minimize the possibility of reporting bias, the data were collected by individuals who were not responsible for the intervention. Second, the level of motivation of the participants to manage their pain was not known. It is possible that the participants in the control group were more motivated to manage their pain than those in the intervention group, and therefore took the initiative to seek out more information after reading the pain management pamphlet. Another limitation is that a relatively small sample size was included in this pilot study, which limits the generalizability of the study’s findings. Attrition bias may exist in this pilot study, affecting the statistical power of the study so that non-statistically significant results were found. Significant differences might have been found with a larger sample size. 

## 8. Conclusions

This is the first RCT to examine the effectiveness of a 12-week peer-led pain education program among nursing home residents. Pain self-efficacy improved significantly after the program. The feedback on the peer-led PAP from nursing home residents was positive. The findings of this study can add to the body of knowledge on pain education delivered by peer volunteers. We are in the process of carrying out the main study, with a larger sample size, to further examine the effectiveness of the peer-led pain education program. 

## Figures and Tables

**Figure 1 ijerph-17-04090-f001:**
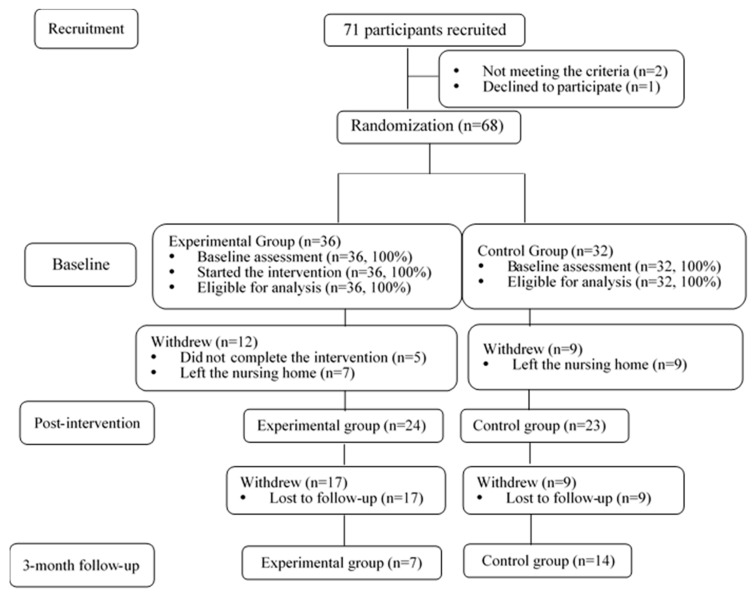
Study flow chart.

**Table 1 ijerph-17-04090-t001:** Demographic characteristics.

Variables	Total (*N* = 68) (*N*, %)	Experimental Group (*n* = 36) (*n*, %)	Control Group (*n* = 32) (*n*, %)	*p*-Value *
**Gender**				0.771
Female	50 (73.5)	27 (75.0)	23 (71.9)	
Male	18 (26.5)	9 (25.0)	9 (28.1)	
**Age group (years)**				0.649
60–70	3 (4.4)	1 (2.8)	2 (6.2)	
71–80	15 (22.1)	6 (16.7)	9 (28.1)	
81–90	33 (48.5)	19 (52.8)	14 (43.8)	
91–100	16 (23.5)	9 (25.0)	7 (21.9)	
**Marital status**				0.293
Single	1 (1.5)	1 (2.8)	0	
Married	18 (26.5)	8 (22.2)	10 (31.3)	
Divorced	2 (2.9)	0	2 (6.3)	
Widowed	45 (66.2)	25 (69.4)	20 (62.5)	
**Education level**				0.939
Uneducated	25 (36.8)	13 (36.1)	13 (40.6)	
Primary school	26 (38.2)	13 (36.1)	13 (40.6)	
Secondary school	15 (22.1)	9 (25.0)	6 (18.8)	
University or above	1 (2.9)	1 (2.8)	1 (3.)	
**Occupation**				0.807
Physical laborer	27 (39.7)	15 (41.7)	12 (37.5)	
Technical job	15 (22.1)	9 (25.0)	6 (18.8)	
Clerk	10 (14.7)	6 (16.7)	4 (12.6)	
Housewife	8 (11.8)	3 (8.3)	5(15.6)	
Others	7 (10.3)	3 (8.3)	4(12.6)	
**Length of institutionalization**				0.531
<1 year	15 (22.1)	8 (22.2)	7 (21.9)	
1–3 years	28 (41.2)	13 (36.1)	15 (37.5)	
4–5 years	7 (10.3)	2 (5.6)	5 (15.6)	
6–10 years	6 (8.8)	4 (11.1)	2 (6.3)	
>10 years	4 (5.9)	3 (8.3)	1 (3.1)	
**Chronic diseases**				
Heart disease	9 (13.4)	4 (11.1)	5 (15.6)	0.648
Diabetes	22 (32.4)	10 (27.8)	12 (37.5)	0.486
Hypertension	27 (39.7)	13 (36.1)	14 (43.8)	0.649
Tracheal disease	2 (2.9)	0	2 (6.3)	0.139
Cataract	16 (23.5)	8 (22.2)	8 (25.0)	0.889
Stroke	7 (10.3)	5 (13.9)	2 (6.3)	0.265
Parkinson disease	1 (1.5)	0	1 (3.1)	0.299
Arthritis	8 (11.8)	2 (5.6)	6 (18.8)	0.109
Physical disability	1 (1.5)	0	1 (3.1)	0.299
Other chronic disease	5 (7.6)	3 (8.3)	2 (6.3)	0.693

* A *p*-value of < 0.05 was considered statistically significant.

**Table 2 ijerph-17-04090-t002:** Pain: experimental group vs. control group over time.

	Group	Experimental Group	Control Group	*Between Group* *p*-Value *	Cohen’s *d^* *(95% CI)*
Time Point					
**Pain self-efficacy**					
**T0**		37.56 ± 13.38	39.67 ± 12.78	0.502	−0.16(−8.26–4.04)
**T1**		41.60 ± 11.55 ^a1^	40.33 ± 12.10 ^b1^	0.674	0.11(−4.64–7.19)
**T2**		46.56 ± 10.81 ^a2^	38.77 ± 5.27 ^b2^	**0.040**	0.92(0.37–15.22)
**Pain intensity**					
**T0**		5.69 ± 3.01	5.63 ± 2.39	0.914	0.02(−1.33–1.19)
**T1**		5.08 ± 3.13 ^a3^	5.43 ± 2.47 ^b3^	0.738	−0.12(−1.27–1.79)
**T2**		5.00 ± 2.83 ^a4^	5.36 ± 3.46 ^b4^	0.471	−0.11(−3.39–1.57)
**Pain interference**					
**T0**		2.63 ± 2.46	2.75 ± 2.07	0.819	−0.05(−1.18–0.94)
**T1**		2.11 ± 2.23 ^a5^	2.48 ± 1.99 ^b5^	0.710	−0.18(−1.32–0.90)
**T2**		2.16 ± 1.68 ^a6^	2.69 ± 2.68 ^b6^	0.731	−0.24(−1.44–2.05)

T0: baseline; T1: post intervention; T2: 3-month follow up. * A *p*-value of < 0.05 was considered statistically significant. ^ Guideline for Cohen’s d: small, d = 0.2; medium, d = 0.5; and large, d = 0.8. a: *p*-values of the within-experimental group comparison. a1: P_T1-T0_ = **0.019**, a2: P_T2-T1_ = **0.002**. a3: P_T1-T0_ = 0.606, a4: P_T2-T1_ = 0.507. a5: P_T1-T0_ = **0.002**, a6: P_T2-T1_ = 0.953. b: *p*-values of the within-control group comparison. b1: P_T1-T0_ = 0.762, b2: P_T2-T1_ = 0.654. b3: P_T1-T0_ = 0.468, b4: P_T2-T1_ = 0.505. b5: P_T1-T0_ = 0.165, b6: P_T2-T1_ = 0.371.

**Table 3 ijerph-17-04090-t003:** Depression: experimental group vs. control group over time.

	Group	Experimental Group	Control Group	Between Group *p*-Value *	Cohen’s *d^* *(95% CI)*
Time Point					
**Geriatric Depression Scale**					
**T0**		4.36 ± 3.10	4.44 ± 2.75	0.913	−0.027(−1.28–1.45)
**T1**		3.50 ± 2.30 ^a7^	4.04 ± 3.27 ^b7^	0.704	−0.191(−1.18–1.74)
**T2**		3.86 ± 1.68 ^a8^	4.21 ± 2.46 ^b8^	0.848	−0.166(−1.43–1.74)

T0: baseline; T1: post intervention; T2: 3-month follow up. * A *p*-value of < 0.05 was considered statistically significant. ^ Guideline for Cohen’s d: small, d = 0.2; medium, d = 0.5; and large, d = 0.8. a: *p*-values of the within-experimental group comparison. a7: P_T1-T0_ = 0.163, a8: P_T2-T1_ = 0.281. b: *p*-values of the within-control group comparison. b7: P_T1-T0_ = 0.100, b8: P_T2-T1_ = 0.379.

**Table 4 ijerph-17-04090-t004:** Pain knowledge: experimental group vs. control group over time.

	Group	Experimental Group	Control Group	*Between Group* *p*-Value *	Cohen’s *d^* *(95% CI)*
Time Point					
**Pain knowledge**					
**T0**		45.45 ± 19.59	44.95 ± 20.61	0.916	0.02(−8.19–9.92)
**T1**		53.41 ± 21.33 ^a9^	51.38 ± 16.13 ^b9^	0.538	0.11(−13.64–7.12)
**T2**		59.74 ± 19.44 ^a10^	55.84 ± 21.91 ^b10^	0.865	0.19(−16.65–19.82)

T0: baseline; T1: post intervention; T2: 3-month follow up. * A *p*-value of < 0.05 was considered statistically significant. ^ Guideline for Cohen’s d: small, d = 0.2; medium, d = 0.5; and large, d = 0.8. a: *p*-values of the within-experimental group comparison. a9: P_T1-T0_ = 0.211, a10: P_T2-T1_ = 0.061. b: *p*-values of the within-control group comparison. b9: P_T1-T0_ = 0.042, b10: P_T2-T1_ = 0.522.

**Table 5 ijerph-17-04090-t005:** Feedback and comments from the older adults.

Categories	Feedback and Comments from Older Adults
**About the program**	I like this program.
	I feel happy and relaxed when taking part in the program every week.
	Some of the contents are really helpful, e.g., exercise.
	I like the massage part most, because I cannot do it by myself.
	The various activities in the program make it interesting.
	Some of the contents are difficult to memorize.
**About the peer volunteers (Experimental group only)**	The volunteers are very patient and nice.
	I like the volunteers.
	I hope the volunteers can come and visit us more often.

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
