# Peer review of "An Exploration of the Effectiveness of a Peer-Led Pain Management Program (PAP) for Nursing Home Residents with Chronic Pain and an Evaluation of Their Experiences: A Pilot Randomized Controlled Trial"

_ijerph, 2020, doi:10.3390/ijerph17114090_

Round 1
Reviewer 1 Report
Thanks for having used a GEE model, more appropriate.
Reviewer 2 Report
Dear Authors,
I have seen an improvement in the article. I consider is ready for publication. Thank you very much for bringing this idea and I do encourage to convert this pilot study in a great research.
Kind regards
This manuscript is a resubmission of an earlier submission. The following is a list of the peer review reports and author responses from that submission.
Round 1
Reviewer 1 Report
Dear Authors,
Your manuscript has been an interest reading. Overall, I like how you have describe your introduction and methods, but results and discussion needs a bit of work.
Methods: Authors have to check the flow-chart, there's text missing in the "3-months follow up, control group"
Results: Some of the results needs to needs to be rewritten. The paragraph about demographic characteristics could improve to make it more interesting for the reader.
Discussion: The authors will ned to focus in the significant differences the found (T1 in the pain intensity). Also, it would be great to determine the sample calculation authors will need to find more consistent data for further investigations.
Discussion needs to be more in depth regarding the literature
Looking forward to hearing from you soon
Regards
Author Response
|
Comments |
Response |
|
1. Methods: Authors have to check the flow-chart, there's text missing in the "3-months follow up, control group" |
Thank you for your valuable comment. We have checked the figure and added the missing text. |
|
2. Results: Some of the results needs to needs to be rewritten. The paragraph about demographic characteristics could improve to make it more interesting for the reader.
|
Thank you for your valuable comment. We have rewritten the results section.
The paragraph on demographic characteristics has been revised as follows: A total of 68 participants who satisfied the criteria were recruited. Thirty-six were allocated to the experimental group and 32 to the control group. The demographic characteristics of the participants are shown in Table 1. More females (73.5%) joined the study than males (26.5%). All of the participants were aged between 60 and 100. More than half of the participants were widows. More than one-third of the participants were uneducated. Forty-one percent of the participants had resided in a nursing home for 1 to 3 years, and 6% for over 10 years. Hypertension was the most commonly reported chronic disease (39.7%). No statistically significant differences in demographic characteristics were found between the experimental and control groups.
|
|
3. Discussion: The authors will need to focus in the significant differences the found (T1 in the pain intensity). Also, it would be great to determine the sample calculation authors will need to find more consistent data for further investigations.
|
Thank you for your valuable comment. We have rewritten the discussion section on pain intensity as follows: There was a significant reduction in pain intensity after the treatment, and the pain intensity score of the experimental group was significantly lower than that of the control group at the post-treatment assessment. This result is consistent with that of several previous peer-led pain education programs, which also reported improvements in pain intensity[8][16][17][18]. This finding is promising, and indicates that our peer-led pain management program is effective in reducing pain.
|
|
4. Discussion needs to be more in depth regarding the literature
|
Thank you for your valuable comment. We have rewritten the discussion section. |

Reviewer 2 Report
Dear authors,
Thanks for submitting your work to the journal. You describe a cluster randomized trial where peer support was used to help individuals manage their chronic pain over a 12-week peer-led. Participant’s satisfaction and acceptance were evaluated by semi-structured interview. You conclude that pain outcomes and satisfaction was better in the intervention group.
The study is interesting, relevant, addressing appropriately an important clinical problem.
However, the conclusions are not supported by the results and the authors should rewrite the results section and not overinterpret their results. This overinterpretation render all the work doubtful, what is pity.
Specifically:
- Pain outcomes: If the ANOVA is not statistically significant, t-test should not be performed
- Differences between the groups and between the time points, if not confirmed statistically, should not be considered at all. Please rewrite the results and the discussion sections accordingly.
- Please considere to use appropriate guidelines (e.g. available on equator-network.org) to write and check your manuscript. Some important information are still missing, for instance regarding the treatment of information in the qualitative part.
Minor comments:
- Please clarify for the reader, including in the figure, what are the reasons for the drop out
- The last step of the control group in the figure (flow chart) does not appears correctly.
Reviewer 3 Report
I appreciate the work you have done and I find it of great interest. Nevertheless, to improve the manuscript I present below some issues and some incongruities that I have found.
Suggestions
- I suggest authors to change in keywords "elderly" for "older adults".
- Incongruence: have a criteria of people age more than 55 years old and choose people from nursing home and selected people aged more than 60.
- Figure 1. The caracters size should be more balanced, there are evident differences between them. This figure is confused. I suggested to change the way it is presented to be more understandable.
- I suggest for future research to add a quality of life test.
Certain questions
- To access the material from Google Drive. How older adults access to this material? Did they need help from nursing home's professionals?
- About "4.2.3. Pain knowledge": How do you assess whether an answer is correct or not? Has a rough answer been valid?
- Why haven't you analyzed the cognitive impairment with the data you have obtained?
- In the discussion: the authors didn't add future improvements. Although this is the first experience with this topic, the discussion has remained poor and needs to be more complete.
Round 2
Reviewer 2 Report
Dear authors,
Thanks for clarifying some points. However, it is disappointing to see that you are still overstating, overinterpreting your results and drawing conclusions simply not supported by the data.
Namely:
- There is a very high risk of attrition bias. Must be mentioned
- There is no strategy for multiple comparisons, multiple testing. Such a strategy would probably lead to the disappearance of statistical significance. But this is science, and you have to accept it.
- You cannot conclude on any difference for the depression scores, none are statistically different, so, not different.
Reviewer 3 Report
I appreciate the incorporation of my suggestions. Good job! I will wait for your future work.
Author Response
Thanks for your comments ad suggestions again.
Round 3
Reviewer 2 Report
Dear authors,
It is very disappointing to see that you are still, despite my efforts, overstating, overinterpreting your results and drawing conclusions simply not supported by the data.
Namely:
- There is a very high risk of attrition bias (=incomplete outcome data). Not attribution bias.
- There is no strategy for multiple comparisons, multiple testing. You should ask the help of an experienced statistician. An ANOVA for comparisons between the groups should probably be used, but some assumptions must then be checked. I think that the only statistically significant difference will then disappear, but this is not a problem per se. Concluding on inappropriate test is a bigger problem.
- You are still concluding on difference for the depression scores, none being statistically different, so, not different, in the abstract and the results section.